# Substrate binding and catalytic mechanism of the *Se*-glycosyltransferase SenB in the biosynthesis of selenoneine

Wei Huang[1,2,3], Jun Song[1,2,3], Tianxue Sun[2], Yue He[2], Xiang Li[1], Zixin Deng [2] & Feng Long [1,2] ✉

Selenium is an essential multifunctional trace element in diverse organisms. The only *Se*-glycosyltransferase identified that catalyzes the incorporation of selenium in selenoneine biosynthesis is SenB from *Variovorax paradoxus*. Although the biochemical function of SenB has been investigated, its substrate specificity, structure, and catalytic mechanism have not been elucidated. Here, we reveal that SenB exhibits sugar donor promiscuity and can utilize six UDP-sugars to generate selenosugars. We report crystal structures of SenB complexed with different UDP-sugars. The key elements N20/T23/E231 contribute to the sugar donor selectivity of SenB. A proposed catalytic mechanism is tested by structure-guided mutagenesis, revealing that SenB yields selenosugars by forming C-Se glycosidic bonds via spontaneous deprotonation and disrupting Se-P bonds by nucleophilic water attack, which is initiated by the critical residue K158. Furthermore, we functionally and structurally characterize two other *Se*-glycosyltransferases, *Cb*SenB from *Comamonadaceae bacterium* and *Rs*SenB from *Ramlibacter* sp., which also exhibit sugar donor promiscuity.

Selenium (Se) is a trace element that was first discovered from sulfur ore in 1817[1], and moderate levels of Se are essential for maintaining normal life activities[2,3]. However, Se deficiency and excess Se are associated with several human diseases[4]. In nature, Se can perform biological roles in inorganic forms, such as sodium selenite, or in organic forms, such as selenocysteine (Sec), which can be incorporated into physiologically active selenoproteins[3]. The Se utilization pathways are different in eukaryotic and prokaryotic systems. Although Sec is commonly carried as selenocysteinylated tRNA^Sec, it is converted from *O*-phosphoserylated tRNA^Sec by the catalysis of Sep-SecS in humans[5]; in contrast, it is produced from serylated tRNA^Sec in bacteria by the catalysis of SelA[6,7]. Other organic forms of Se, including selenouridine (SeU), which is synthesized by SelU and used to modify microbial nucleic acids[8,9], and cofactors for some redox enzymes, such as molybdenum dehydrogenase[10,11], have been identified in bacteria. In

addition, selenoneine (SEN), a Se analog of ergothioneine, was found in various sea-dwelling animals and in β-proteobacteria[12-15]. With the exception that SEN could protect against mercury toxicity and act as an effective antioxidant[14,16,17], little was known about its other physiological roles.

The Se-specific biosynthetic pathways have been limited exclusively to Sec and SeU[5-9]. A specific SEN biosynthetic pathway was identified in *Variovorax paradoxus* DSM 30034, which is recognized as a way to incorporate Se into organic matter in bacteria[15]. The bacterial SEN biosynthesis gene cluster encodes three proteins, SenA, SenB, and SenC. SenB catalyzes the generation of the key intermediate selenoglucose product using SeP synthesized by selenophosphate synthetase (named SelD or SenC)[15] (Fig. 1). SenA is an Fe-dependent SEN synthase that catalyzes the formation of Se–C bonds between hercynine and selenosugars[15]. SenB was identified as the first reported

---

[1]Department of Neurosurgery, Zhongnan Hospital of Wuhan University, School of Pharmaceutical Sciences, Wuhan University, Wuhan 430071, China. [2]Ministry of Education Key Laboratory of Combinatorial Biosynthesis and Drug Discovery, School of Pharmaceutical Sciences, Wuhan University, Wuhan 430071, China. [3]These authors contributed equally: Wei Huang, Jun Song. ✉e-mail: longfe@whu.edu.cn

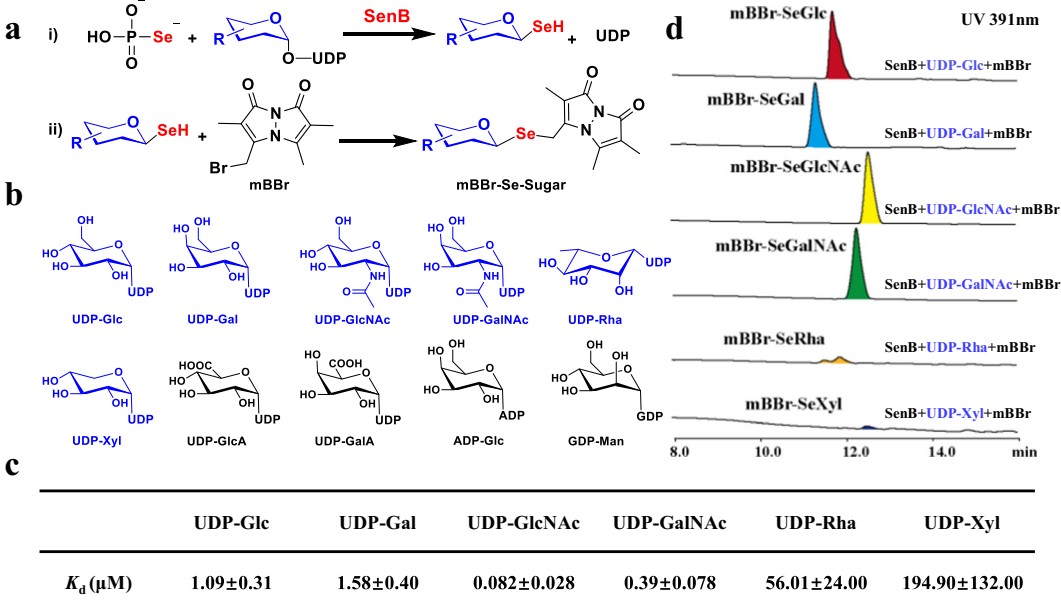

**Fig. 1 | Bacterial SEN biosynthetic pathway.** SenB catalyzes generation of the key intermediate selenoglucose product using SeP synthesized by SenC.

**Fig. 2 | Probing sugar donor selectivity of SenB. a,** i) SenB catalyzes the synthesis of selenosugars using SeP; ii) Derivatization of selenosugars using mBBr. **b** Chemical structures of sugar donors. **c** $K_d$ values of SenB for binding of different sugar donors. **d** HPLC-UV/DAD analysis at 391 nm of the mBBr derivatives of the SenB reaction products using different sugar donors.

| | UDP-Glc | UDP-Gal | UDP-GlcNAc | UDP-GalNAc | UDP-Rha | UDP-Xyl |
|---|---|---|---|---|---|---|
| $K_d$ (μM) | 1.09±0.31 | 1.58±0.40 | 0.082±0.028 | 0.39±0.078 | 56.01±24.00 | 194.90±132.00 |

selenosugar synthase[15]. The catalytic function of SenB was demonstrated to be distinct from that of other known Se−C bond-forming enzymes because SenB can utilize UDP-Glc, UDP-GlcNAc, or UDP-GalNAc to catalyze the generation of corresponding selenosugars in the absence of cofactors[5−7,9,15]. The reaction catalyzed by SenB is more similar to the catalytic process of glycosyltransferases (GTs), suggesting that SenB could also be considered as a *Se*-glycosyltransferase (*Se*GT).

GTs are a class of enzymes that transfer sugar groups from sugar donors to sugar acceptors. GTs are widespread in living organisms and are involved in the glycosylation of primary and secondary metabolites[18−20]. GTs can be further categorized by their various three-dimensional structures (GT-A, GT-B, GT-C, and GT-D) or by the different glycosidic bonds formed in the products (O-/N-/S-/C-GTs)[18,21]. SenB is the only *Se*GT reported thus far. In contrast to other types of GTs, SenB performs dual functions in two-step catalytic reactions, catalyzing the formation of C−Se glycosidic bonds and the subsequent cleavage of Se-P bonds to generate the final product[15]. Since the catalytic function and sequence similarity of SenB are low compared to those of other reported enzymes, the structure and catalytic mechanism of SenB remain unclear and should be investigated in-depth.

In this work, we perform structure–function analysis and deduce the catalytic mechanism of SenB through functional characterization, crystallization, and structure-based mutagenesis. Moreover, we mine and identify two other *Se*GTs, *Cb*SenB and *Rs*SenB, the functional and structural findings of which strongly support our proposed catalytic mechanisms that lead to sugar donor promiscuity and *Se*-glycosylation of *Se*GTs. Through these studies, we provide the structure of *Se*GTs and insights into the diversity of C−Se bond formation as well as Se−P bond cleavage in nature.

## Results and discussion

### Probing sugar donor selectivity of SenB

Previous studies have shown that SenB can utilize UDP-GlcNAc, UDP-GalNAc, and UDP-Glc as sugar donors to generate corresponding selenosugars[15]. UDP-GlcNAc and UDP-GalNAc are a class of sugar donors with large spatial dimensions due to the presence of an *N*-acetyl group on the C-2′ of the sugar moiety. Therefore, SenB may also utilize other types of sugar donors. We selected ten different sugar donors, namely, UDP-Glc, UDP-Gal, UDP-GlcNAc, UDP-GalNAc, UDP-Rha, UDP-Xyl, UDP-GlcA, UDP-GalA, ADP-Glc, and GDP-Man, and utilized SenC-produced SeP as a sugar acceptor to test the sugar donor specificity of SenB (Fig. 2a, b and Supplementary Fig. 1). The monobromobimane (mBBr) derivatives with a single Se atom were obtained from the final product of the enzymatic reaction using the thiol-labeling reagent and were detected by HPLC-DAD/MS. The results showed that SenB exhibits sugar donor promiscuity (Fig. 2d and Supplementary Fig. 2). In addition to the three sugar donors UDP-Glc, UDP-GlcNAc and UDP-GalNAc reported in the literature[15], SenB can utilize three other UDP-sugars, namely, UDP-Gal, UDP-Rha and UDP-Xyl. SenB exhibits specificity for the UDP form of the sugar donor, and

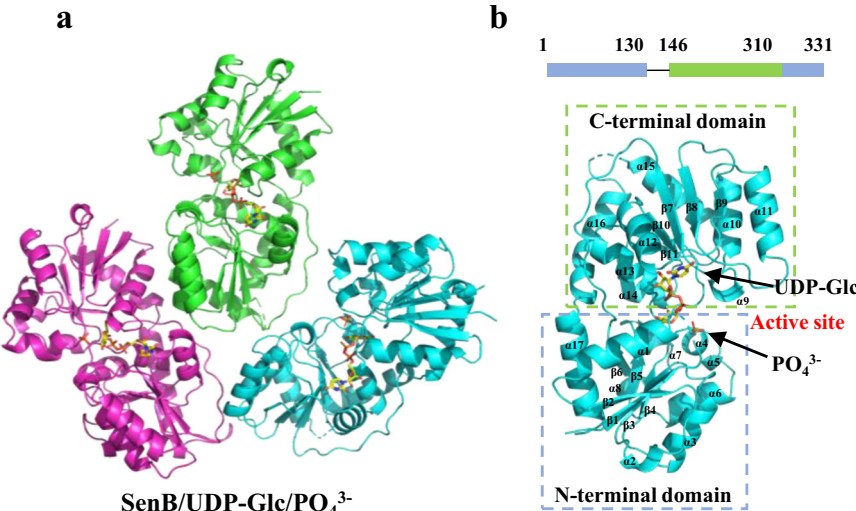

**a**

**SenB/UDP-Glc/PO$_4^{3-}$**

**b**

C-terminal domain

UDP-Glc

Active site

PO$_4^{3-}$

N-terminal domain

**Fig. 3 | Overall structure of SenB. a** Cartoon diagram of the three copies of the ternary SenB/UDP-Glc/PO$_4^{3-}$ complex within one asymmetric unit in crystal. **b** Cartoon diagram of the monomeric ternary SenB/UDP-Glc/PO$_4^{3-}$. The N-terminal domain of SenB is enclosed with a green box and the C-terminal domain with a blue box. The α-helix and β-strand secondary structures are labeled in black font. UDP-Glc and PO$_4^{3-}$ are indicated by sticks.

in addition to the GDP form of the sugar donor that is not recognized by SenB, as reported previously[15], the ADP form of the sugar donor is also not utilized by SenB. Compared with those of UDP-Glc, UDP-Gal, UDP-GlcNAc and UDP-GalNAc, the activity of SenB was lower towards UDP-Rha and UDP-Xyl. The dissociation constants ($K_d$) of SenB for different sugar donors varied from nM to μM (Fig. 2c and Supplementary Fig. 3). The preference of SenB is as follows: UDP-GlcNAc > UDP-GalNAc > UDP-Glc > UDP-Gal > UDP-Rha > UDP-Xyl. This finding is consistent with the results of previous substrate competition studies (UDP-GlcNAc > UDP-GalNAc > UDP-Glc)[15]. Given the chemical structure and catalytic activity of the sugar donor, these findings suggest that the *N*-acetyl group of C-2′ and the hydroxymethyl group of C-5′ on the sugar moiety contribute to the catalytic efficiency of SenB. Selenosugars, which exhibit a variety of promising biological activities, are obtained mainly by chemical syntheses and involve disadvantageous limitations, such as complex reaction conditions, poor yields and selectivity, and lack of diversity[22–24]. The sugar donor promiscuity of SenB could alleviate the drawbacks of chemically synthesized selenosugars, and SenB likely serves as a promising and potentially applicable enzyme for the efficient and green synthesis of selenosugars with different structures.

## Overall crystal structure of SenB

To elucidate the structural basis leading to substrate recognition and the catalytic mechanism of SenB, we solved the structures of SenB complexed with various sugar donors, including the ternary complex SenB/UDP-Glc/PO$_4^{3-}$ (1.95 Å) and two binary complexes, SenB/UDP-GlcNAc (1.88 Å) and SenB/UDP-GalNAc (1.64 Å) (Fig. 3a, Supplementary Fig. 4 and Supplementary Table 1). In all solved crystal structures, each asymmetric unit contains three copies of SenB, which are unlikely to be functionally related due to the lack of protein–protein interactions between them. These findings correspond with the size-exclusion chromatography results in which SenB was demonstrated to be monomeric in solution (Supplementary Fig. 1a). The structure of monomeric SenB consists of two domains that contain a Rossmann-like fold, the N-terminal domain (NTD; residues 1–130, 311–331) and the C-terminal domain (CTD; residues 146–310), which are connected by a loop (residues 131–145). The NTD contains 6 parallel β-folds and 9 α-helices, whereas the CTD contains 5 parallel β-folds and 8 α-helices. The active site of SenB is present in a narrow cleft formed by the face-to-face apposition of the NTD and CTD, with UDP-sugar bound to the

CTD and PO$_4^{3-}$ bound to the NTD (Fig. 3b). These structural features are similar to those of the GT-B type of GTs[18]; thus, SenB is likely a GT-B glycosyltransferase. According to the structural similarity analysis performed using the DALI server[25], the two hits most similar to those of SenB were a sucrose synthase (PDB ID: 6KIH) from *Thermosynechococcus vestitus*[26], with an RMSD (root mean square deviation) of 2.4 Å for Cα atoms and 18% sequence similarity, and a GT-B GT BshA (PDB ID: 6N1X) from *Staphylococcus aureus*[27,] with an RMSD of 2.9 Å for Cα atoms and 12% sequence similarity. This finding suggested that the catalytic mechanism of SenB may differ significantly from that of the reported enzymes.

## Structural mechanisms for sugar donor binding and promiscuity of SenB

Structural superposition of SenB/UDP-Glc, SenB/UDP-GlcNAc and SenB/UDP-GalNAc indicated that the spatial positions of the UDP moieties of the three sugar donors overlapped well (Supplementary Fig. 5a, b). The interaction between SenB and UDP was examined in detail (Supplementary Fig. 5c and Supplementary Table 2). Alanine mutagenesis screening revealed that mutations in the amino acids that directly interacted with different moieties of UDP affected the catalytic activity of SenB at various levels. For instance, the catalytic activity of SenB was nearly eliminated by single mutations of the residues that interacted with the phosphate moiety (K158A) or the ribose moiety (E239A). The catalytic activity of the L209A and T214A mutants related to the uracil interaction increased by approximately 1.5-fold (Supplementary Fig. 5d).

In contrast to most GT-B-type GTs, which have known structures that mainly use π-π interactions to stabilize uracil[28–32], SenB binds uracil by forming numerous hydrogen bonds. A substantial space limitation around uracil was observed in SenB. The enhanced catalytic activity of the mutants L209A and T214A likely results from reduced spatial hindrance. Nevertheless, the compact UDP pocket of SenB cannot easily accommodate nitrogenous bases with large molecular backbones, such as adenosine (A) and guanosine (G) (Supplementary Fig. 6). This could explain why SenB utilized only UDP-sugars but not ADP-sugars or GDP-sugars.

In our solved complex structures of SenB, the electron densities of the three different sugar moieties of the UDP-sugars were well defined (Supplementary Fig. 4). To reveal the structural mechanism that underlies the sugar donor promiscuity of SenB, models of SenB in

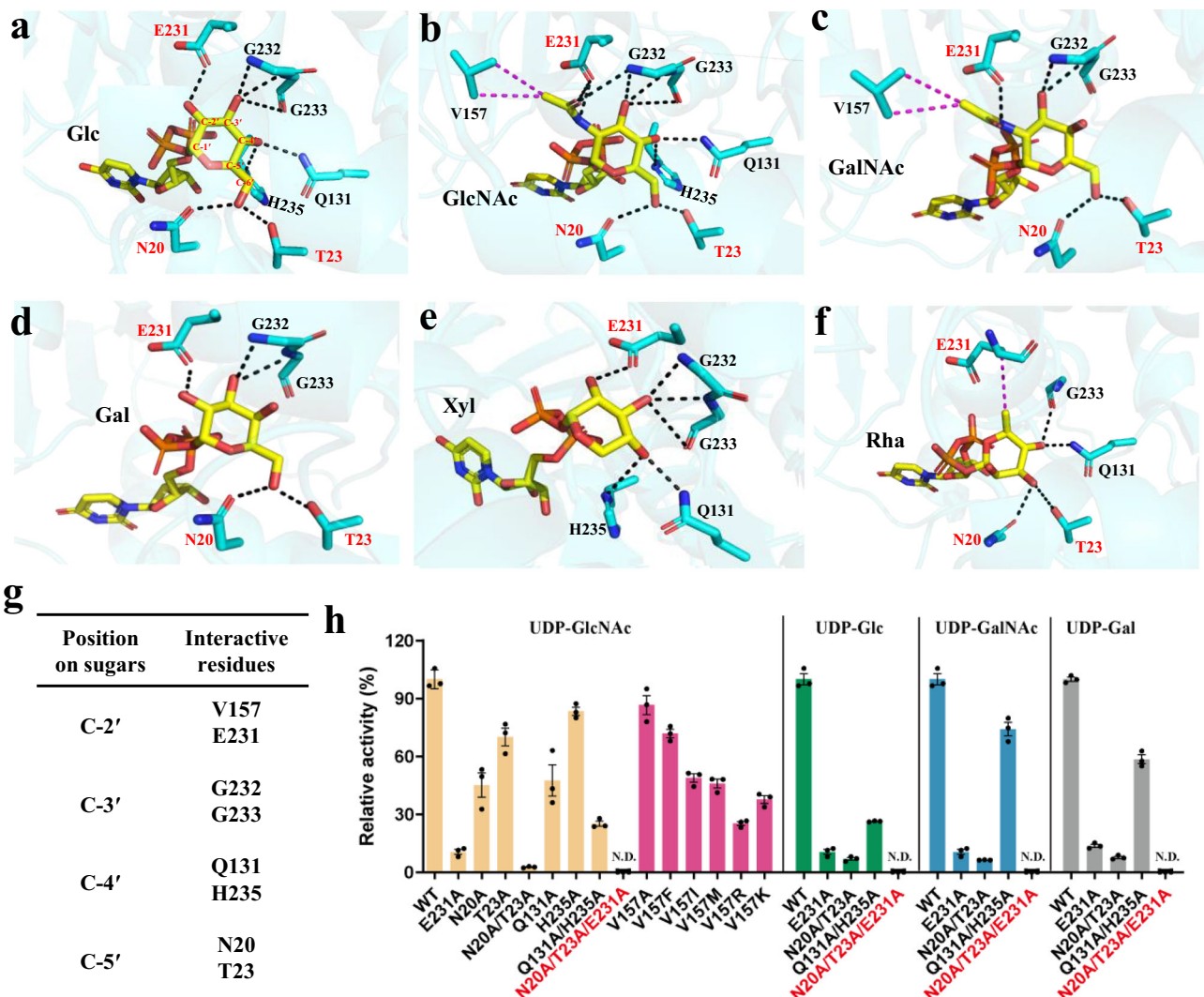

**Fig. 4 | Structural mechanisms for sugar donor binding and promiscuity of SenB. a–f** Interactions between the sugar moieties of different sugar donors and SenB. Hydrogen bonds are indicated by black dashed lines and hydrophobic interactions are indicated by magenta dashed lines. **g** Summary of residues that interact with different sites of the sugar donor. **h** Relative catalytic activities of the SenB mutants using different UDP-sugar donors. Data are presented as mean values ± SD (*n* = 3 independent experiments). Source data are provided as a Source Data file.

complex with other UDP-sugars, including SenB/UDP-Gal, SenB/UDP-Xyl and SenB/UDP-Rha, were further constructed by docking based on the crystal structure of SenB/UDP-Glc/PO₄³⁻. An in-depth comparative analysis of the interaction between the sugar moiety and SenB was performed (Fig. 4a–g). Since the chemical groups at the C-2′ and C-5′ positions of the sugar moiety are correlated with the catalytic efficiency of SenB, the residues that interact with these chemical groups likely influence the catalytic activity of SenB. We next compared interactions involving the C-2′ position of the sugar moiety. E231 of SenB tightly interacts with the C-2′ groups of sugars in all six complex structures. Therefore, it was speculated that E231 is also important for the catalytic activity of SenB. The mutagenesis results showed that the relative catalytic activities of the E231A mutant for different UDP-sugars were significantly lower (<20%) (Fig. 4h). The acetyl groups in UDP-GlcNAc and UDP-GalNAc may enhance the affinity of SenB for additional hydrophobic interactions with V157 compared to that of other UDP-sugars. Weakening this interaction by altering V157 to various amino acids (A/F/I/M/R/K) decreased the activity of all mutants, yet these enzymes retained some catalytic function (Fig. 4h). Comparative analysis revealed that N20 and T23 were essential for stabilizing the C-5′ hydroxymethyl group of the sugar moiety in each

complex. Although only moderate decreases in relative catalytic activity were observed for the N20A or T23A single mutants, the catalytic activity of the N20A/T23A double mutant decreased to less than 10% in the presence of different sugar donors (Fig. 4h). These results suggested that N20 and T23 are also catalytically important residues of SenB for different sugar donors. Further combinatorial mutagenesis showed that the relative catalytic activity of SenB was completely lost after the triad mutant N20A/T23A/E231A was introduced for all tested sugar donors (Fig. 4h). These findings indicate that the N20/T23/E231 site is crucial for the strong preference of SenB for certain sugar donors. Residues interacting with other sites of the sugar moiety do not affect the sugar donor promiscuity of SenB, although the residues can affect the catalytic activity of SenB. For instance, the catalytic activities were partially preserved but slightly varied with different sugar donors in the single and double mutants of Q131 and H235, the residues of which interact with C-4′ of the sugar moiety (Fig. 4h).

**Structural basis for SeP binding and the catalytic mechanism of SenB**

To investigate the catalytic mechanism of SenB, the structural basis of SeP binding must be clarified. SenB/UDP-Glc/SeP, a ternary complex

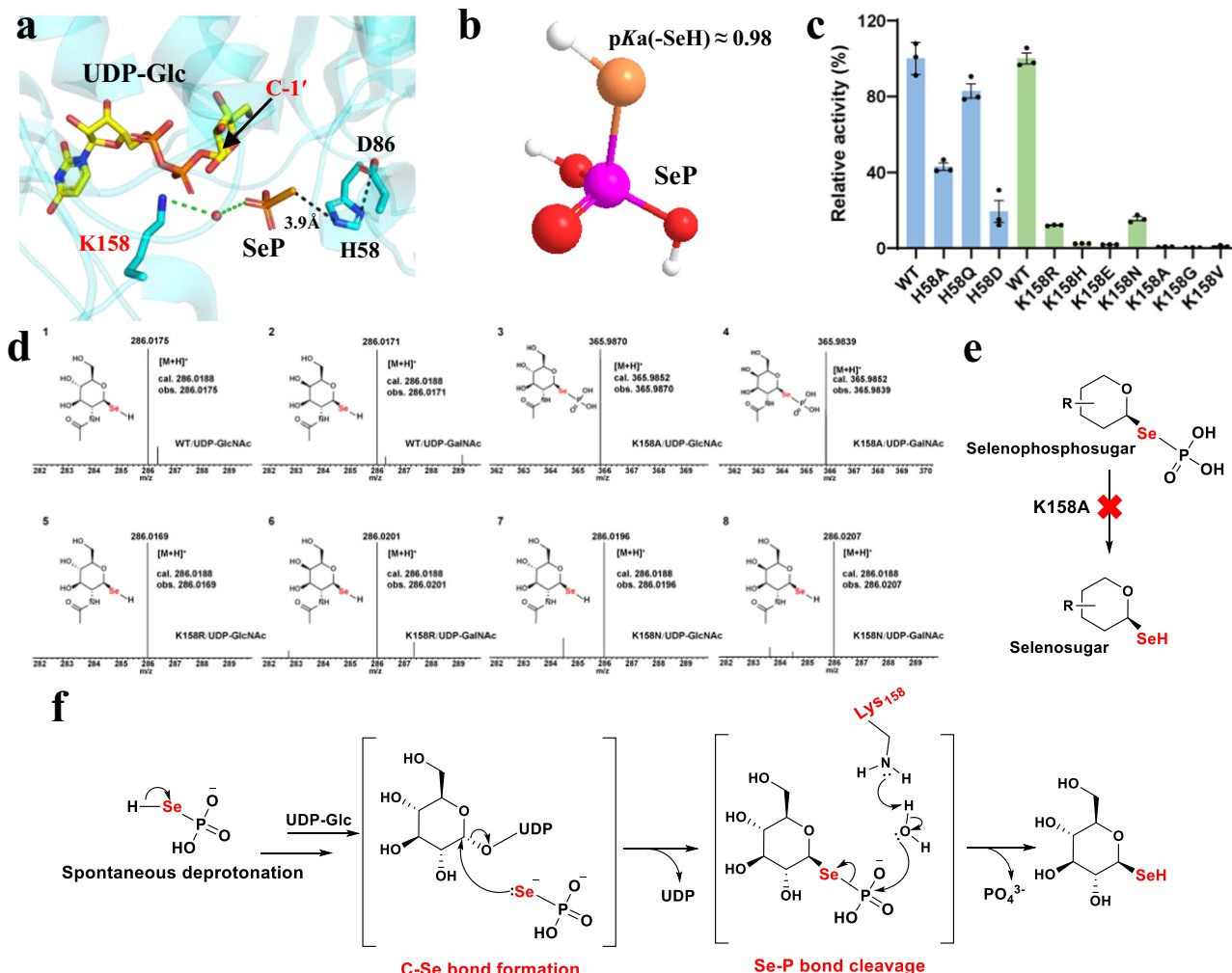

**Fig. 5 | Catalytic mechanism of SenB. a** Catalytic activity center and critical catalytic residues of SenB. Hydrogen bonds are represented by black dashed lines, water bridges by green dashed lines, and water molecules by red spheres. **b** The p$K$a value of the selenol group (-SeH) of SeP. **c** Relative catalytic activities of the SenB mutants using UDP-Glc as the sugar donor. Data are presented as mean values ± SD ($n$ = 3 independent experiments). Source data are provided as a Source Data file. **d** LC-MS detection of the selenophosphosugar intermediates from catalysis by the K158 mutants. **e** The K158A mutant halts the reaction of the Se-P bond cleavage. **f** Proposed reaction mechanism for the *Se*-glycosylation of SenB.

model that represents the reaction state, was constructed by docking based on the structure of SenB/UDP-Glc/PO₄³⁻ (Supplementary Fig. 7a, d). SeP was shown to bind in a narrow pocket near UDP-Glc, consisting of a series of hydrophilic residues (N20, H58, R61, T83, T85, and R155), which stabilize SeP in the pocket by forming abundant hydrogen bonds (Supplementary Fig. 7b, c). Single alanine mutations of the above hydrophilic residues hampered the activities of SenB. However, when we performed multipoint mutations to further disrupt the pocket hydrophilicity, the N20A/T85A, T83A/T85A and N20A/T83A/T85A mutations resulted in a complete loss of activity (Supplementary Fig. 7e). Based on these results, the hydrophilicity of the sugar acceptor binding pocket ensures the stable binding of SeP and is essential for the catalytic activity of SenB.

The formation of Se−C glycosidic bonds is the first step in the production of selenosugars by *Se*GTs. SenB could generate 1-*Se*-β-D-glucose from UDP-α-D-glucose[15], suggesting that the configuration at the anomeric carbon of the sugar moiety is flipped during the catalytic process. Thus, the generation of Se−C glycosidic bonds by SenB possibly involves the deprotonation of Se in the presence of a catalytic base and direct nucleophilic attack on C-1′ of the UDP-sugar (Fig. 5a). Deprotonation of the acceptor is important during the process of glycosidic bond formation by most GTs. In GT-B-type GTs, many plant natural product UGTs, as well as a few microbial UGTs, contain a conserved catalytic dyad His-Asp that accounts for acceptor deprotonation[18,33–36] (Supplementary Fig. 8). In the His-Asp-dependent GT-B-type GT structures, the spatial localization of the His-Asp catalytic dyad was almost identical, as both were closely clamped by the UDP-sugar and acceptor[18,33–36] (Supplementary Fig. 9). Although the corresponding catalytic dyad (His58-Asp86) of SenB was found in the neighborhood of the SeP-binding site (Fig. 5a), structural localization revealed that this pair of residues in SenB is located at the outer edge of the SeP acceptor and away from the UDP-sugar, which has not been observed in other structures of GT-B-type GTs. Moreover, our mutagenesis results showed only a reduction but not complete loss of enzymatic activity in the H58A, H58Q, and H58D mutants (Fig. 5c). The protein folding of these three purified mutants was demonstrated to be identical to that of the wild type by size-exclusion chromatography analysis, excluding the possibility that the reduced catalytic activity was caused by structural misfolding of SenB (Supplementary Fig. 10). These results suggest that the deprotonation of SeP is not dependent on His58-Asp86, implying that the His-Asp in SenB is not a catalytic dyad and that SenB may utilize a different strategy to deprotonate the acceptor. The selenol group (-SeH) of SeP is deprotonated at pH 7.2 since its p$K$a was estimated to be 0.98, suggesting that SeP could be

deprotonated regardless of enzymatic catalysis under our experimental conditions[37] (Fig. 5b). Therefore, deprotonation of SeP during Se–C glycosidic bond formation is likely spontaneous and may not require the assistance of a catalytic base.

SenB necessitates cleavage of the Se-P bond during the catalysis of selenosugar formation. Previously characterized enzymes, such as SelA and SepSecS, have been reported to require the PLP cofactor and specific lysine residues (K258 in SelA and K284 in SepSecS) for assisting in the catalytic process of cleaving Se–P bonds[5,6]. However, SenB lacks the PLP cofactor, suggesting an alternative mechanism for Se-P bond cleavage. Comparative analyses of the SeP binding pockets of SenB, SelA, and SepSecS revealed that a spatially equivalent lysine residue, K158, was present despite the absence of the PLP cofactor in SenB (Fig. 5a). Hence, K158 may serve as the critical basic residue that facilitates Se–P bond cleavage in SenB. Prior structural results indicated that K158 can form a weak hydrogen bond with the β-phosphate of UDP at 3.2 Å (Supplementary Fig. 5c), potentially impacting its ability to abstract protons and hampering its likelihood of acting as a catalytic base. However, a nearly complete loss of catalytic activity was observed for the K158A mutant, suggesting that the hydrogen bond on K158's proton abstraction has a subtle influence and underscoring the significance of K158 for the enzymatic activity of SenB (Supplementary Fig. 5d). Moreover, in the SenB/UDP-Glc/PO$_4^{3-}$ complex structure, K158 forms a stable water-mediated bridge with PO$_4^{3-}$, a phenomenon also observed between SeP and K258 in SelA[6] (Fig. 5a). Thus, the Se-P bond cleavage mediated by K158 in SenB may begin with the amino side chain of K158, which captures a proton from a water molecule. The resulting negatively charged water molecules attack the partially positively charged phosphorus atom in the Se–P bond, leading to bond disruption and the formation of a selenosugar product. Although R155, another basic amino acid near the active pocket of SenB, could hypothetically play a similar role to K158, electron density analyses revealed a rigid conformation for the side chain of K158; in contrast, the electron density for the side chain of R155 was missing or very poor, indicating its extreme flexibility in configuration (Supplementary Fig. 11). These finding, together with the knowledge that the R155A mutant retains substantial catalytic activity, suggest that R155 does not likely act as the catalytic base in SenB (Supplementary Fig. 5d). To further explore the role of K158 in Se-P bond cleavage, the catalytic products of the functionally impaired K158A mutant were analyzed. Using UDP-GlcNAc and UDP-GalNAc as sugar donors, the corresponding selenophosphate sugar intermediates catalyzed by the K158A mutant were detected through LC–MS (Fig. 5d). These findings are consistent with our hypothesis that K158A retains an ability to form Se–C glycosidic bonds but cannot cleave Se-P bonds (Fig. 5e). To test this hypothesis, K158 was mutated to different amino acids, during which most mutants completely lost catalytic activity. Only those with basic or amino side chains, such as the K158R, K158H, and K158N mutants, preserved minimal catalytic activity (Fig. 5c). No selenophosphate sugar intermediates were detected in the weakened mutants K158R or K158N, in which only trace amounts of the final selenosugar product were observed (Fig. 5d). Taken together, these findings demonstrate that SenB catalyzes the formation of C–Se glycosidic bonds and the cleavage of Se–P bonds through a mechanism that involves spontaneous deprotonation and the potential catalytic residue K158 (Fig. 5f), which is distinct from the catalytic mechanisms used by other GTs reported to date.

## Mining and characterization of SenB-like enzymes

We performed gene mining for homologous *Se*GTs in the NCBI database using SenB as a probe, and more than 200 SenB-like genes were found (>55% similarity), which were mainly distributed in β-proteobacteria. Sequence conservation analysis of these SenB-like enzymes revealed that the putative catalytic residue K158 exhibits a high degree of conservation (Supplementary Fig. 12), which further

supports our hypothesis regarding the catalytic mechanism of SenB. In addition, a highly conserved "EGGAHV" motif related to sugar donor binding was found in SenB-like enzymes (Fig. 4g and Supplementary Fig. 12). One of the three key amino acids that determines the sugar donor promiscuity of SenB, E231, is located within this conserved motif. The other two key amino acids, N20/T23, are also highly conserved among these SenB-like enzymes. Therefore, these SenB-like enzymes may exhibit a certain degree of sugar donor promiscuity. We selected two SenB-like enzymes, *Cb*SenB (GenBank: RYF17368.1; 65.5% sequence similarity), which was derived from *Comamonadaceae bacterium*, and *Rs*SenB (GenBank: MBC7468551.1, 65.2% sequence similarity), which was derived from *Ramlibacter* sp., for further functional characterization (Supplementary Fig. 1c–f and Supplementary Table 3). Like SenB, *Cb*SenB and *Rs*SenB are located in the SEN biosynthesis-related gene cluster (Supplementary Fig. 13). Sequence alignment of *Cb*SenB and *Rs*SenB revealed the putative catalytic residue K158 as well as the key EGGAHV motif (Fig. 6a). Catalytic activity analysis revealed that *Cb*SenB and *Rs*SenB recognize sugar donors like SenB and utilize UDP-Glc, UDP-GlcNAc, UDP-Gal, and UDP-GalNAc to generate the corresponding selenosugars (Fig. 6b–e). The catalytic activities of these three enzymes for UDP-Glc, UDP-GlcNAc, and UDP-Gal were similar, but *Rs*SenB utilized UDP-GalNAc more strongly than SenB and *Cb*SenB did. Mutagenesis assays confirmed that the residues equivalent to N20/T23/E231 of SenB are critical for acceptance of various sugar donors in *Cb*SenB and *Rs*SenB (Supplementary Fig. 14). To further confirm our hypothesis on the catalytic mechanism of *Se* glycosylation, we solved the crystal structure of *Rs*SenB (PDB ID: 8K5U) in the apo form at a resolution of 2.15 Å (Supplementary Table 4). Unlike that of SenB, the asymmetric unit of the crystal contains two *Rs*SenB molecules (Fig. 6f). The monomer structure of *Rs*SenB is very similar to that of SenB, with an RMSD of 0.99 Å for Cα atoms. The spatial positions of the putative catalytic residue K158 in *Rs*SenB strongly overlapped with the structures of SenB (Fig. 6g). In addition, the mutagenesis results for the catalytic residues in *Cb*SenB and *Rs*SenB were consistent with those for SenB (Fig. 6h). These results not only provided two other *Se*GTs but also further demonstrated the universal effects of the key amino acids N20/T23/E231 on the promiscuity of sugar donors and the putative catalytic residue K158 on the *Se* glycosylation of SenB.

In conclusion, SenB is the only functionally identified *Se*GT in nature, but its structure and catalytic mechanism were unknown. In this work, substrate specificity investigations showed that SenB exhibits sugar donor promiscuity and specificity for the utilization of six sugar donors in the form of UDP. The structures of SenB in complex with three sugar donors were solved, namely, SenB/UDP-Glc/PO$_4^{3-}$, SenB/UDP-GlcNAc, and SenB/UDP-GalNAc. Structural analysis revealed that the loss of π-π interactions, leading to increased steric hindrance near uracil, is the major factor in the specific recognition of UDP-form sugar donors by SenB. Structural comparison and mutagenesis revealed the critical amino acids N20, T23, and E231 that modulate the sugar donor preference of SenB. Furthermore, catalytic mechanistic investigations revealed that SenB may undergo spontaneous deprotonation to form a C-Se bond and that the Se-P bond may cleave via a putative catalytic residue, K158. Finally, two other *Se*GTs, *Rs*SenB and *Cb*SenB, were functionally and structurally characterized. These results revealed the structure of SenB and the possible mechanism of *Se* glycosylation, providing insights into the diversity of C–Se bond formation as well as Se-P bond cleavage in nature. In addition, the results provide theoretical guidance for structure-based engineering modifications of *Se*GT for selenosugars or Se-related drug synthesis.

## Methods
### Materials and reagents
All reagents were purchased from Sigma-Aldrich (St. Louis, MO, USA) unless otherwise specified. mBBr, UDP-Glc, UDP-GlcNAc, UDP-Gal,

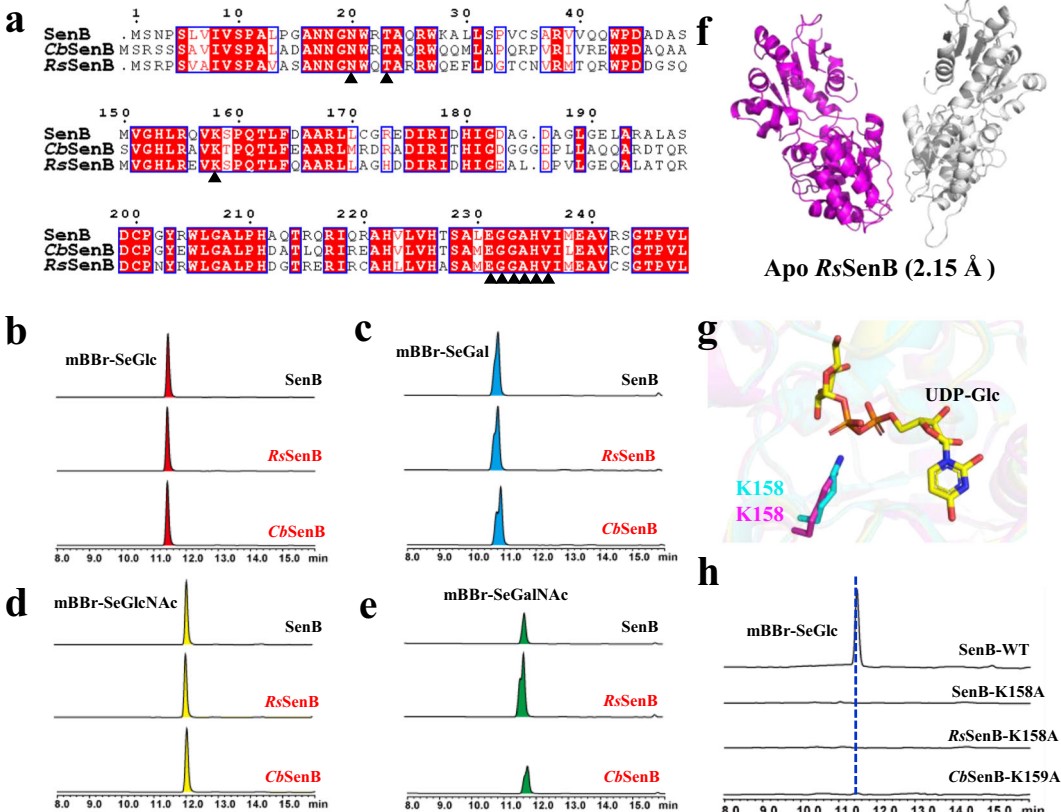

**Fig. 6 | Characterization of SenB-like enzymes. a** Protein sequence comparison of *Cb*SenB, *Rs*SenB and SenB. N20/T23, K158 and the EGGAHV motif are labeled with black triangles. **b–e** HPLC-UV/DAD analysis of the catalytic activities of *Cb*SenB and *Rs*SenB using four different UDP-sugars. **f** Overall structure of *Rs*SenB. Two molecules within one asymmetric unit are shown. **g** Spatial positions of K158 in SenB (cyan) and *Rs*SenB (magenta). **h** HPLC-UV/DAD analysis of the catalytic activities of putative catalytic residue mutants of *Rs*SenB and *Cb*SenB using UDP-Glc as the sugar donor.

UDP-GalNAc, UDP-Xyl, UDP-Rha, UDP-GlcA, GDP-Man and ADP-Glc were purchased from Meryer Biotechnology (Shanghai, China). Cloning Phanta Max Super-Fidelity DNA Polymerase was purchased from Vazyme Biotechnology (Nanjing, China). Hieff Clone Plus Multi One Step Cloning Kit was purchased from Yeasen Biotechnology (Shanghai, China). Restriction Enzymes was purchased from New England BioLabs (Ipswich, MA, USA). Codon-optimized gene fragments were purchased from Tsingke (Beijing, China). All crystallization materials and reagents were purchased from Hampton Research (Laguna Niguel, CA, USA). Acetonitrile and formic acid of HPLC grade were purchased from Thermo Fisher Scientific (Waltham, MA, USA).

## Plasmid construction for protein expression

All protein coding DNA sequences were codon-optimized for expression in *Escherichia coli*, and commercially synthesized. The DNA fragments were then amplified using Super-Fidelity DNA Polymerase, and ligated with the *Nde*I and *Hin*dIII linearized pET28a vector using one step cloning kit following the manufacturer's instructions. Ligation mixtures were transformed into chemically competent *E. coli* TOP10 by heat shock and plated onto LB agar containing 50 μg/mL kanamycin. Single colony was picked and cultured for plasmid extraction. The constructions confirmed by Sanger sequencing were later used for protein expression.

## Site-directed mutagenesis

Site-directed mutagenesis of SenB was performed using the polymerase chain reaction (PCR) with primers designed to generate the desired mutations (Supplementary Table 5). The wild-type pET28a-SenB plasmid was used as the PCR template. PCR was set up with a Phanta Max Super-Fidelity DNA Polymerase in a Biorad C1000 Thermal Cycler. In a 25 μL of reaction (12.5 μL of reaction buffer, 50 ng of plasmid template, 125 ng of forward primer, 125 ng of forward primer, 1 μL of dNTP, and 1 μL of polymerase), the mixture was placed in a thermal cycler following a traditional three-step PCR protocol (Initial denaturation at 95 °C for 30 s, 25 cycles of denaturation at 95 °C for 30 s, annealing at 58 °C for 30 s and extension at 72 °C for 200 s, final extension at 72 °C for 10 min and hold in 4 °C). The template DNA was removed from the reaction by the *Dpn*I digestion at 37 °C for 2-h. The digestion products were then transformed into the *E. coli* TOP10 competent cells. The plasmids extracted from the transformants were sequenced to verify existence of the desired mutations.

## Expression and purification of SenB, SenC, *Cb*SenB, *Rs*SenB and mutants

All proteins were individually produced in *E. coli* Rosetta (DE3) cells. Cells transformed with corresponding expression plasmids were cultured in 1 L of LB medium (1% Tryptone, 0.5% Yeast extract, 1% NaCl) containing 50 μg/mL kanamycin, shaken at 220 rpm and 37 °C until the $OD_{600}$ reached 0.6, then induced with 0.4 mM isopropyl β-D-1-thiogalactopyranoside (IPTG) and growth at 18 °C for 18 h. Cells were pelleted by centrifugation (SORVALL LYNX 4000, Thermo Scientific) at $6000 \times g$ for 10 min, resuspended in 80 mL of lysis buffer consisting of 20 mM Tris-HCl, 300 mM NaCl, pH 7.5, and disrupted by a high-pressure homogenizer (EmulsiFlex-C3, AVESTIN, Canada) at 12,000 psi. The lysate was clarified by centrifugation at $15,000 \times g$ for 45 min, and the supernatant was loaded onto a gravity column pre-equilibrated in lysis buffer with 5 mL Ni-NTA affinity resins (GenScript, Nanjing, China). The column was washed sequentially with lysis buffer containing 20 mM imidazole for 10 CV (column volume), 50 mM imidazole for 6 CV, and then the target proteins were eluted with lysis

buffer containing 300 mM imidazole for 3 CV. The eluted proteins were further purified by size-exclusion chromatography using a HiLoad 16/600 Superdex 75 column (GE Healthcare) in a buffer containing 20 mM Tris-HCl pH 7.5, 150 mM NaCl, and 1 mM dithiothreitol (DTT). The peak fractions containing target protein were collected and examined by 10% sodium dodecyl sulfate-polyacrylamide gel electrophoresis (SDS-PAGE). Finally, the purified protein was concentrated to 7.5 mg/mL using an Amicon Ultra-30 K filter (Millipore), flash-frozen in the liquid nitrogen, and stored at −80 °C for later use.

### Enzyme activity assay of SenB, *Rs*SenB, *Cb*SenB and mutants
Under anaerobic conditions, the reactions were performed in a final volume of 50 µL, containing 20 µM SenC, 20 µM SenB, *Rs*SenB, *Cb*SenB or mutants, 2 mM DTT, 2 mM ATP, 1 mM $Na_2Se$, and 2 mM of UDP-sugar. All of materials were prepared in buffer consisting of 50 mM Tris-HCl, 20 mM KCl and 5 mM $MgCl_2$, pH 7.2. Experimental reactions were prepared in an identical manner, except that different mutants of SenB and UDP-sugar were used. After a 6-h incubation period at room temperature, reactions were removed from the glovebox and exposed to atmosphere for 30 min to oxidize any unreacted $Na_2Se$. The reactions were then quenched with 50 µL of 10 mM ice cold mBBr in MeCN, followed by incubation in dark at room temperature for an additional 30 min to allow the completeness of derivatization with mBBr. Finally, the supernatants were collected after centrifugation at 12,000 × *g* for 30 min, and analyzed by HPLC-UV/DAD and HPLC-MS.

### HPLC and LC−MS analysis
The HPLC analysis was performed on a Shimadzu-LC-20AT (Japan) with an Ultimate® XB-C18 column (4.6 mm × 250 mm I.D., 5 µm, Welch Materials, Inc., China) at a flow rate of 0.8 mL/min, using the mobile phase of (A) 0.1% formic acid in deionized $H_2O$ and (B) 100% MeCN. The gradient settings for separating the products and substrates were 0−20 min 10% B to 50% B, 20−25 min 50% B to 100% B, 25−28 min 100% B to 10% B, and 28−35 min 10% B to 10% B. The products were further confirmed using an LTQ XL Orbitrap mass spectrometer (Thermo Fisher Scientific Inc.) The MS/MS analysis was carried out in a positive ionization mode with 35% relative collision energy. The relative activities of the mutants were determined by HPLC and calculated by the product's peak area dividing the wild-type's peak area. All experiments were performed in triplicate.

### Microscale thermophoresis assay
The purified SenB protein was exchanged into test buffer consisting of 25 mM Hepes, 150 mM NaCl, 0.05% Tween 20, 1 mM DTT, pH 7.5 for the microscale thermophoresis (MST) experiments. The protein was diluted to a final concentration of 1 µM in test buffer, then a 100 nM MO-L018 RED-tris-NTA dye solution was added to protein solution. The protein and dye mixture were mixed well and incubated at 4 °C in the dark for 30 min. The labeled protein was obtained by centrifugation at 12,000 x g for 10 min. The binding affinities between substrates and proteins were analyzed on a Monolith NT.115 instrument (Nanotemper Technologies). Different concentrations of UDP-sugars were serially diluted from the stocks using test buffer (the premade stocks were 40 µM UDP-Glc, 50 µM UDP-Gal, 10 µM UDP-GlcNAc, 20 µM UDP-GalNAc, 1 mM UDP-Rha, and 10 mM UDP-Xyl, respectively). The equal volumes of labeled protein were added to various concentrations of UDP-sugar solutions in a final volume of 10 µL. After being incubated at 4 °C in the dark for 30 min, the reaction mixtures were loaded into standard treated capillaries (Monolith NT.115 series capillaries MO-K022) and analyzed by MST at medium MST power and auto-detect excitation power with a laser-on time of 2.5 s and a detection temperature of 25 °C. The $K_d$ values were calculated using MO. Affinity Analysis Software from three independent thermophoresis measurements.

### Selenophosphosugar intermediate analysis
Under anaerobic conditions, enzyme activity assays of the K158A mutant of SenB were performed in an identical manner, except being supplied with different UDP-sugars (UDP-Glc/UDP-GlcNAc/UDP-GalNAc). After a 6-h incubation period at room temperature, reactions were removed from the glovebox and exposed to atmosphere for 30 min to oxidize any unreacted $Na_2Se$. The reactions were then quenched with 50 µL MeCN, and centrifuged at 12,000 × *g* for 30 min. The supernatants were analyzed by LC−MS. The LC was performed with an COSMOSIL® PBr column (4.6 mm×250 mm I.D.) at a flow rate of 0.3 mL/min, using the mobile phase of (A) 10 mM $NH_4Ac$ in deionized $H_2O$ and (B) 100% MeCN. The Elution settings were 0 − 25 min 5% B. The products were further confirmed using an LTQ Orbitrap Elite mass spectrometer (Thermo Fisher Scientific Inc.)

### Protein crystallization
The purified SenB protein (7.5 mg/mL) was incubated with UDP-Glc (5 mM), UDP-GlcNAc (5 mM), or UDP-GalNAc (5 mM) for 30 min on ice, and the purified *Rs*SenB protein (7.5 mg/mL) was incubated with UDP-GalNAc (5 mM), before set-up of the crystallization trays. Crystals of SenB/UDP-Glc/$PO_4^{3-}$ were observed at 18 °C within 3-4 d using the hanging drop vapor diffusion method by mixing 0.8 µL of protein with 0.8 µL of the reservoir solution (0.2 M Ammonium sulfate, 0.1 M Hepes (pH 7.5), and 20% (w/v) polyethylene glycol 8000, 10% (v/v) 2-Propanol). The crystals of SenB/UDP-GalNAc were obtained in the reservoir solution containing 0.1 M Bis-Tris (pH 6.5), and 20% (w/v) polyethylene glycol 5000-MME. The crystals of SenB/UDP-GlcNAc were obtained in the reservoir solution containing 0.1 M Hepes (pH 7.0), and 15% (w/v) polyethylene glycol 20000. The crystals of *Rs*SenB/UDP-GalNAc were obtained in the reservoir solution containing 0.2 M Ammonium citrate dibasic, and 20% (w/v) polyethylene glycol 3350. All crystals were harvested in the same reservoir solution supplemented with 20% (w/v) glycerol as the cryo-protectant and flash-frozen in the liquid nitrogen.

### Data collection and structure determination
The crystallographic data sets were collected on the beamlines 19U1 at the Shanghai Synchrotron Radiation Facility[38]. The diffraction images were processed using XDS[39]. The structures of SenB and *Rs*SenB were solved by molecular replacement using a Phaser from the CCP4 suite[40], and the alphafold2[41] predicted structure of SenB was used as the searching model. The models of the SenB complexes were built initially using AutoBuild[42] and manually using Coot[43]. The iterative refinement and structure validation were done using Phenix[44].

### Structure analysis
Structural visualization analysis and figure preparation were made with Protein-ligand interaction profiler[45] and PyMOL (The PyMOL Molecular Graphics System, Version 2.0 Schrödinger, LLC). Sequence alignments were created using Clustal Omega[46], ESPript[47], and WebLogo[48].

### Molecular docking
Autodock 4.0[49] was used to build the structures of SenB/UDP-Gal, SenB/UDP-Xyl, SenB/UDP-Rha, and SenB/UDP-Glc/SeP, using the structure of SenB/UDP-Glc/$PO_4^{3-}$ as the template.

### Reporting summary
Further information on research design is available in the Nature Portfolio Reporting Summary linked to this article.

## Data availability
The atomic coordinates and structure factors of SenB and *Rs*SenB generated in this study have been deposited in the Protein Data Bank (www.rcsb.org) under accession codes 8JJN (SenB/UDP-Glc/$PO_4^{3-}$ structure), 8JJT (SenB/UDP-GlcNAc structure), 8JJQ (SenB/UDP-GalNAc

structure) and 8K5U (*Rs*SenB structure). Source data are provided with this paper.

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

## Acknowledgements

We thank the staffs from BL19U1 beamlines of National Facility for Protein Science in Shanghai (NFPS) at Shanghai Synchrotron Radiation Facility for their assistance during data collection, and Prof B.Y from Yangzhou University for providing selenoglucose as a control. This work was supported by grants from the National Key Research and Development Program of China (Grant No. 2021YFA0909500 to F.L.), the National Natural Science Foundation of China (Grant No. 82304333 to W.H.), the Fundamental Research Funds for the Central Universities of China (Grant No. 2042019kf0185 to F.L.), the Wuhan University Zhongnan Hospital Science and Technology Innovation Cultivation Fund (Grant No. CXPY2023011 to W.H.) and the Translational Medicine and Inter-disciplinary Research Joint Fund of Zhongnan Hospital of Wuhan University (Grant No. ZNJC202245 to F.L.).

## Author contributions

W.H. and F.L. designed the experiments. J.S., W.H. and T.S. performed the in vitro enzymatic analysis. W.H., J.S. and Y.H. performed crystallization experiments. W.H., J.S. and F.L. analyzed the data. W.H. drafted the manuscript. X.L., Z.D. and F.L. revised the manuscript.

## Competing interests

The authors declare no competing interests.
