## [Peer Review File · Nature Communications]

Substrate binding and catalytic mechanism of the Se-glycosyltransferase SenB in the biosynthesis of selenoneineEditorial Note: This manuscript has been previously reviewed at another journal that is not operating a transparent peer review scheme. This document only contains reviewer comments and rebuttal letters for versions considered at *Nature Communications*.

Reviewer #1 (Remarks to the Author):

This is a revised submission of a paper that describes the biochemical and structural characterization of a novel glycosyltransferase involved in a three gene cluster that catalyzes the synthesis of a Se-containing metabolite in a broad collection of bacteria. The initial submission had numerous concerns regarding the writing, summary of prior work on selenium metabolism, and interpretation of the experimental data. The revised submission addressed some of these concerns, but numerous problems in the writing in the revision result in continuing overall problems with the manuscript. These major and minor concerns include numerous problems with English usage and interpretation of data.

The editing and data interpretation concerns include:

Abstract, line 20: 'tested' not 'testified'

Abstract, lines 21-22: "...structure-guided mutagenesis indicating that SenB forms C-Se glycosidic bonds via spontaneous deprotonation and nucleophilic attack with the critical residue K158 assisting in Se-P bond cleavage..."

Abstract, line 23: "Furthermore, we functionally and structurally..."

Introduction, line 34: 'majorly' is not appropriate word usage here.

Introduction, line 38: I am not sure that 'catalyzation' is a word.

Introduction, line 44: 'With the exception that SEN could protect against...'

Introduction, lines 47-49: The sentence starting with 'Recently...' is very awkwardly worded. Please revise.

Introduction, line 59: '...glycosyltransferases (GTs), suggesting that SenB could also be considered as a Se-glycosyltransferase...'

Results and discussion, line 81: 'Previous studies showed that SenB can utilize UDP-GlcNAc, UDP-GalNAc, and UDP-Glc as sugar donors...'

Results and discussion, line 102: 'This result is consistent with...'

Results and discussion, line 105: '...on the sugar contribute to the catalytic efficiency...'

Results and discussion, line 150: This paragraph describes the structural basis for the specificity toward uridine versus adenine/guanine bases for the sugar nucleotide donor, but then attempts to use the K158A and E239A mutations as justification. K158 interacts with the beta phosphate and E239 interacts with the ribose. Neither interact with the nucleotide base to provide an explanation for base specificity.

Results and discussion, line 153-154: The sentence that starts with 'Limited...' makes no sense.

Results and discussion, line 161: 'It was proposed earlier...' Where was it proposed earlier? Is there a reference?

Results and discussion, line 174: ‘...hydroxymethyl group of the sugar moiety in each complex.’

Results and discussion, line 181-182: ‘These results suggested that the core site composed of N20/T23/E231 determines the sugar donor promiscuity of SenB.’ I’m not sure that I understand the logic here. These residues interact with all gluco- and galacto-configured sugars similarly because the C2 and C6 hydroxyls of these sugars are all bound in structurally equivalent positions. They confer binding affinity to all four of the sugars at C2OH and C6OH. The ‘promiscuity’ comes from the fact that there is not a steric barrier or profound binding affinity difference for the gluco- versus galacto-epimers at C4, so UDP derivatives of Glc(±NAC) and Gal(±NAC) are all ‘allowed’. I am not sure that I agree that “N20/T23/E231 determines the sugar donor promiscuity of SenB”.

Results and discussion, line 209-210: “...GT-B type GTs, a highly conserved catalytic dyad His-Asp mainly accounts for acceptor deprotonation^{18,36-38}” This is a considerable overstatement. The His-Asp dyad is only conserved for a minor subset of GT-B enzymes (mostly plant natural product UDP-glycosyltransferases).

Results and discussion, line 216-217: “catalytic dyad His-Asp is almost identical, both being closely sandwiched by the UDP-sugar and acceptor^{18,36-38}” Based on Fig. 5a, the His-Asp is not sandwiched by the UDP-sugar and acceptor. Besides, in the SenB structure in Fig. 5a D86 is not even facing H58.

Results and discussion, line 240-243: “However, the near complete loss of catalytic activity of the K158A mutant was observed, suggesting that the hydrogen bond's influence on K158's proton abstraction may be minimal, and it underscores the significance of K158 for the enzymatic activity of SenB.” I am not sure that I understand the logic here. If the role of K158 is to stabilize the forming negative charge on the beta phosphate during sugar transfer (with no role as catalytic base in cleavage of Se-PO₄) then you would expect the K158A mutation to inactivate the enzyme.

Results and discussion, line 275: ‘...K158 exhibits...’

Results and discussion, line 302: ‘...(Fig. 6h)...’

Results and discussion, line 307: ...structure and catalytic mechanism were completely unknown.”

Results and discussion, line 314-315: “...and mutagenesis revealed the critical amino acids N20, T23, and E231 that control sugar donor promiscuity of SenB.” I still don’t agree that they ‘control’ promiscuity.

Overall, the manuscript is improved in some areas, but the writing and interpretation of some of the data is still problematic. Please consider revisions to the above sections.

Reviewer #2 (Remarks to the Author):

Huang and co-workers reported the cocrystal structures of SenB, a Se-glycosyltransferase and investigated the enzyme’s recognition and catalytic mechanism. By performing biochemical studies using different sugars, the authors characterized SenB’s substrate promiscuity toward these ligands. Through follow-up structural modeling, they identified the key residues for substrate binding and catalysis, and corroborated the results by enzymatic activity assays. The combined structural and biochemical studies provided insights into the reaction mechanism of SenB and contributed to our understanding on the selenium transfer mechanism as well. Overall, the authors did a good job on the biochemical and structural studies, and they carefully proposed a tentative mechanism for SenB catalysis. While the results of the manuscript and the high-quality structural work are interesting, the authors need to clarify several following issues before the manuscript is accepted.

1. While proposing the mechanism (line 248 on page 10), the authors say “SenB involves the amino

side chain of K158 capturing a proton from a water molecule, becoming a negatively charged nucleophile that attacks the positively charged phosphorus atom in the Se-P bond...". There are problems with this sentence and the related figure (Fig. 5f). First, it was the water molecule that becomes negatively charged, not SenB; second, the phosphorus atom here is only "partially positively charged" due to the unequal polarities between P and Se; third, there are several problems with in the "C-Se bond formation" and "Se-P cleavage" steps of the Fig. 5f cartoon, where the arrows were pointing to the wrong atoms or positions;

2. Site-Directed Mutagenesis in the Methods section: the quikchange method should be clearly stated;

3. The manuscript could've used some language editing; typos or grammatical errors are seen. For instance,

a. line 222 on page 9 "...deploy different strategy": missing "a"

b. line 379 on page 15 "...Tri-HCl"?

c. line 381 on page 15 "...a 6 h incubation period": missing a hyphen;

Reviewer #3 (Remarks to the Author):

I already wrote in my previous review of this paper about the relevance and value of this work; I stand by it: it's a valuable addition for selenium research. Regarding the revision:

The minor issues I raised were addressed.

I still recommend improvements in English language throughout the whole document; perhaps authors can use a professional service? Or the journal can step in to help?

Point-by-point response

REVIEWER COMMENTS

Reviewer #1 (Remarks to the Author):

This is a revised submission of a paper that describes the biochemical and structural characterization of a novel glycosyltransferase involved in a three gene cluster that catalyzes the synthesis of a Se-containing metabolite in a broad collection of bacteria. The initial submission had numerous concerns regarding the writing, summary of prior work on selenium metabolism, and interpretation of the experimental data. The revised submission addressed some of these concerns, but numerous problems in the writing in the revision result in continuing overall problems with the manuscript. These major and minor concerns include numerous problems with English usage and interpretation of data.

Response:

We greatly appreciate the reviewer for valuable comments and corrections. We have corrected the English usage and adjusted the interpretation of our data according to the comments. The language has been further improved from the Nature Research Editing Service.

The editing and data interpretation concerns include:

Abstract, line 20: ‘tested’ not ‘testified’

Response:

The “testified” has been corrected to “tested”.

Abstract, lines 21-22: “...structure-guided mutagenesis indicating that SenB forms C-Se glycosidic bonds via spontaneous deprotonation and nucleophilic attack with the critical residue K158 assisting in Se-P bond cleavage...”

Response:

We have made the corresponding revision. “A proposed catalytic mechanism was tested by structure-guided mutagenesis, revealing that SenB yields selenosugars by forming C-Se glycosidic bonds via spontaneous deprotonation and disrupting Se-P bonds by nucleophilic water attack, which is initiated by the critical residue K158.”

Abstract, line 23: “Furthermore, we functionally and structurally...”

Response:

We have made the corresponding revision. “Furthermore, we functionally and structurally characterized two other Se-glycosyltransferases,”

Introduction, line 34: ‘majorly’ is not appropriate word usage here.

Response:

We have made the corresponding revision. “In nature, Se can perform biological roles in inorganic forms, such as sodium selenite, or in organic forms, such as

selenocysteine (Sec), which can be incorporated into physiologically active selenoproteins.”

Introduction, line 38: I am not sure that ‘catalyzation’ is a word.

Response:

The “catalyzation” has been corrected to “catalysis”.

Introduction, line 44: ‘With the exception that SEN could protect against...’

Response:

We have made the corresponding revision. “With the exception that SEN could protect against mercury toxicity and act as an effective antioxidant,”

Introduction, lines 47-49: The sentence starting with ‘Recently...’ is very awkwardly worded. Please revise.

Response:

We have made the corresponding revision. “A specific SEN biosynthetic pathway was identified in *Variovorax paradoxus* DSM 30034, which is recognized as a novel way to incorporate Se into organic matter in bacteria¹⁵.”

Introduction, line 59: ‘...glycosyltransferases (GTs), suggesting that SenB could also be considered as a Se-glycosyltransferase...’

Response:

We have made the corresponding correction. “...glycosyltransferases (GTs), suggesting that SenB could also be considered as a *Se*-glycosyltransferase (*SeGT*).”

Results and discussion, line 81: ‘Previous studies showed that SenB can utilize UDP-GlcNAc, UDP-GalNAc, and UDP-Glc as sugar donors...’

Response:

We have made the corresponding correction. “Previous studies have shown that SenB can utilize UDP-GlcNAc, UDP-GalNAc, and UDP-Glc as sugar donors to generate corresponding selenosugars¹⁵.”

Results and discussion, line 102: ‘This result is consistent with...’

Response:

We have made the corresponding correction. “This finding is consistent with the results of previous substrate competition studies (UDP-GlcNAc > UDP-GalNAc > UDP-Glc)¹⁵.”

Results and discussion, line 105: ‘...on the sugar contribute to the catalytic efficiency...’

Response:

We have made the corresponding revision. “...on the sugar contribute to the catalytic efficiency of SenB.”

Results and discussion, line 150: This paragraph describes the structural basis for the

specificity toward uridine versus adenine/guanine bases for the sugar nucleotide donor, but then attempts to use the K158A and E239A mutations as justification. K158 interacts with the beta phosphate and E239 interacts with the ribose. Neither interact with the nucleotide base to provide an explanation for base specificity.

Response:

We are sorry for our misleading interpretation. The catalytic activity assay for the representative mutants K158A, E239A, L209A, and T214 was designed to investigate the potential impact of the residues interacting with different parts of UDP on the catalytic activity of SenB. The above listed four residues interact with ribose, phosphate or uracil moieties of UDP. The corresponding mutants showed differently influenced catalytic activity. In our study, the uracil base preference was found mainly associated with the space limitation of the UDP binding pocket. To clarify our results and avoid potential confusion for the readers, we have reorganized the Supplementary Table 2 and rephrased the text on page 6-7, line 143-156.

Supplementary Table 2. Detailed interactions between SenB and UDP.

Residue	Residue atom	Ligand moiety	Ligand atom	Distance (Å)
Hydrogen bond				
R155	NH2	β-phosphate	O2B	2.7
K158	NZ		O2B	3.2
G19	N		O1B	3.2
H235	ND1/N	α-phosphate	O1A	2.8/2.8
V236	N		O2A	3.3
R22	NH1/NE	Ribose ring	O3C	3.0/4.0
V236	OE2		O3C	2.6
E239	OE1		O2C	2.9
N17	ND2	Uracil ring	O2	2.8
L209	O/N		N3/O4	3.2/3.5
T214	OG1		N3	3.3
hydrophobic interaction				
V151	CG1	Uracil ring	C5	3.7

Results and discussion, line 153-154: The sentence that starts with ‘Limited...’ makes no sense.

Response:

The referred sentence has been deleted in the revised manuscript.

Results and discussion, line 161: ‘It was proposed earlier...’ Where was it proposed earlier? Is there a reference?

Response:

We are sorry for our misleading interpretation. We were meant to refer to our finding described on page 5, line 106-108, “the *N*-acetyl group of C-2’ and the hydroxymethyl group of C-5’ on the sugar moiety contribute to the catalytic efficiency of SenB.”. To improve the interpretation, the corresponding sentence has been rephrased as, “Since the chemical groups at the C-2’ and C-5’ positions of the sugar

moiety are correlated with the catalytic efficiency of SenB, the residues that interact with these chemical groups likely influence the catalytic activity of SenB.”

Results and discussion, line 174: ‘...hydroxymethyl group of the sugar moiety in each complex.’

Response:

We have made the corresponding revision. “Comparative analysis revealed that N20 and T23 were essential for stabilizing the C-5’ hydroxymethyl group of the sugar moiety in each complex.”

Results and discussion, line 181-182: ‘These results suggested that the core site composed of N20/T23/E231 determines the sugar donor promiscuity of SenB.’ I’m not sure that I understand the logic here. These residues interact with all gluco- and galacto-configured sugars similarly because the C2 and C6 hydroxyls of these sugars are all bound in structurally equivalent positions. They confer binding affinity to all four of the sugars at C2OH and C6OH. The ‘promiscuity’ comes from the fact that there is not a steric barrier or profound binding affinity difference for the gluco- versus galacto-epimers at C4, so UDP derivatives of Glc(±NAc) and Gal(±NAc) are all ‘allowed’. I am not sure that I agree that “N20/T23/E231 determines the sugar donor promiscuity of SenB”.

Response:

Thank you for your valuable comments. We have carefully considered your concerns regarding the role of residues N20/T23/E231 in determining the sugar donor promiscuity of SenB, and reanalyzed our data accordingly. We agree that our previous statement that the core site composed of N20/T23/E231 determines the sugar donor promiscuity of SenB, was inaccurate. We apologize for this overstatement. However, we still believe that the N20/T23/E231 residues are crucial for both the sugar donor preference and catalytic activity of SenB. As you have pointed out there is no steric hindrance or profound binding environment around the C-4 position, it might allow various sugars. In fact, our results showed the similar C-4 hydroxyl configurations (such as UDP-GlcA and UDP-GalA, with variations at the C-6 positions) could not assure acceptance by SenB as the sugar donor. Preference of sugar donors greatly depends on the interaction strength between UDP-sugar and SenB, which is determined by the interactive residues. Our amended mutation assays also confirmed the impact of N20/T23/E231 on the sugar donor preference of the homologs *CbSenB* and *RsSenB*. Therefore, the N20/T23/E231 residues are essentially involved in the sugar donor preference of SenB. The corrected description and new data (**Figure S14**) have been added in the revised manuscript on page 8. line 185-186 and page 12, line 298-300.

Supplementary Figure 14. HPLC-UV/DAD analysis of the mBBBr derivatives of the reaction products catalyzed by *CbSenB*, *RsSenB* and their mutants using different sugar donors. a-d, Product detection of *CbSenB* and its triple mutant N21A/T24A/E233A using UDP-Glc (a), UDP-Gal (b), UDP-GlcNAc (c), or UDP-GalNAc (d) as the sugar donor. e-f, Product detection of *RsSenB* and its triple mutant N20A/T23A/E231A using UDP-Glc (e), UDP-Gal (f), UDP-GlcNAc (g), or UDP-GalNAc (h) as the sugar donor.

Results and discussion, line 209-210: "...GT-B type GTs, a highly conserved catalytic dyad His-Asp mainly accounts for acceptor deprotonation^{18,36-38}" This is a

considerable overstatement. The His-Asp dyad is only conserved for a minor subset of GT-B enzymes (mostly plant natural product UDP-glycosyltransferases).

Response:

Thank you for your valuable comments. We have corrected the sentence to “In GT-B-type GTs, many plant natural product UGTs, as well as a few microbial UGTs, contain a conserved catalytic dyad His-Asp that accounts for acceptor deprotonation^{18,33-36}.”

Results and discussion, line 216-217: “catalytic dyad His-Asp is almost identical, both being closely sandwiched by the UDP-sugar and acceptor^{18,36-38}” Based on Fig. 5a, the His-Asp is not sandwiched by the UDP-sugar and acceptor. Besides, in the SenB structure in Fig. 5a D86 is not even facing H58.

Response:

We are sorry for our misleading interpretation. Thank you for your valuable comments. For better metaphorical illustration, we changed “sandwiched” to “clamped”. Indeed, the His-Asp in SenB is not clamped between UDP-sugars and the receptor. As mentioned in your comment, D86 is not facing H58. This structural feature actually distinguishes SenB from the other GT-B type glycosyltransferases that depend on a His-Asp catalytic dyad for catalytic function. As shown in Fig.S9, the His-Asp dyad is clamped between the sugar donor and acceptor in many GT-B GTs except SenB. Therefore, the catalytic mechanism of SenB might be unique. Our mutagenesis study indicates that the H58A mutation does not lead to complete inactivation of SenB, implying that SeP's deprotonation does not rely on H58. It could be deduced that the His58-Asp86 in SenB may not function as a catalytic dyad. We apologize for the confusion and have carefully revised the interpretation of our results to enhance the readability. The corresponding changes are shown in the revised manuscript on page 9, line 211-228.

“In GT-B-type GTs, many plant natural product UGTs, as well as a few microbial UGTs, contain a conserved catalytic dyad His-Asp that accounts for acceptor deprotonation^{18,33-36} (**Fig. S8**). In the His-Asp-dependent GT-B-type GT structures, the spatial localization of the His-Asp catalytic dyad was almost identical, as both were closely clamped by the UDP-sugar and acceptor^{18,33-36} (**Fig. S9**). Although the corresponding catalytic dyad (His58-Asp86) of SenB was found in the neighbourhood of the SeP-binding site (**Fig. 5a**), structural localization revealed that this pair of residues in SenB is located at the outer edge of the SeP acceptor and away from the UDP-sugar, which has not been observed in other structures of GT-B-type GTs. Moreover, our mutagenesis results showed only a reduction but not complete loss of enzymatic activity in the H58A, H58Q, and H58D mutants (**Fig. 5c**). The protein folding of these three purified mutants was demonstrated to be identical to that of the wild type by size-exclusion chromatography analysis, excluding the possibility that the reduced catalytic activity was caused by structural misfolding of SenB (**Fig. S10**). These results suggest that the deprotonation of SeP is not dependent on His58-Asp86, implying that the His-Asp in SenB is not a catalytic dyad and that SenB may utilize a different strategy to deprotonate the acceptor.”

Supplementary Figure 9. The spatial localization of the His-Asp in the structures of SenB and other GT-B type GTs.

Results and discussion, line 240-243: “However, the near complete loss of catalytic activity of the K158A mutant was observed, suggesting that the hydrogen bond's influence on K158's proton abstraction may be minimal, and it underscores the significance of K158 for the enzymatic activity of SenB.” I am not sure that I understand the logic here. If the role of K158 is to stabilize the forming negative charge on the beta phosphate during sugar transfer (with no role as catalytic base in cleavage of Se-PO₄) then you would expect the K158A mutation to inactivate the enzyme.

Response:

Thank you for your valuable suggestions. We have changed “minimal” to “subtle” on page 10, line 248. Structural analysis revealed that, in addition to K158 (3.2 Å), residues R155 (2.7 Å) and G19 (3.2 Å) also interact with the β-phosphate of UDP-Sugar in SenB. During glycosyl transfer, all these three residues could stabilize the negative charge on the β-phosphate. To this extent, R155A and K158A might have similar influence on the catalytic activity. However, compared to R155A (~60% relative catalytic activity), the near-complete loss of catalytic activity was observed in the K158A mutant, suggesting that K158 might be involved in other important catalytic process beyond merely stabilizing the negative charge on the β-phosphate. In fact, we found out that K158 is very critical for the Se-P bond cleavage. The amino side chain of K158 captures a proton from a water molecule. The resulted negatively charged water molecule attacks the partially positively charged phosphorus atom in the Se-P bond, and leads to bond disruption and the formation of a selenosugar product. Accordingly, we have amended our interpretation on the SenB catalytic mechanism on page 10, line 249-253.

Results and discussion, line 275: ‘...K158 exhibits...’

Response:

We have made the corresponding correction.

Results and discussion, line 302: ‘...(Fig. 6h)...’

Response:

We have made the corresponding correction.

Results and discussion, line 307: ‘...structure and catalytic mechanism were completely unknown.’

Response:

We have made the corresponding correction. “In conclusion, SenB is the only functionally identified SeGT in nature, but its structure and catalytic mechanism were unknown.”

Results and discussion, line 314-315: “...and mutagenesis revealed the critical amino acids N20, T23, and E231 that control sugar donor promiscuity of SenB.” I still don’t agree that they ‘control’ promiscuity.

Response:

Thank you for your valuable comments and suggestions. We have changed “control sugar donor promiscuity” to “modulate sugar donor preference” in hopes of more accurately describing the functional impact of these amino acids of SenB.

Overall, the manuscript is improved in some areas, but the writing and interpretation of some of the data is still problematic. Please consider revisions to the above sections.

Reviewer #2 (Remarks to the Author):

Huang and co-workers reported the cocrystal structures of SenB, a Se-glycosyltransferase and investigated the enzyme's recognition and catalytic mechanism. By performing biochemical studies using different sugars, the authors characterized SenB's substrate promiscuity toward these ligands. Through follow-up structural modeling, they identified the key residues for substrate binding and catalysis, and corroborated the results by enzymatic activity assays. The combined structural and biochemical studies provided insights into the reaction mechanism of SenB and contributed to our understanding on the selenium transfer mechanism as well. Overall, the authors did a good job on the biochemical and structural studies, and they carefully proposed a tentative mechanism for SenB catalysis. While the results of the manuscript and the high-quality structural work are interesting, the authors need to clarify several following issues before the manuscript is accepted.

Response:

Thank you for your valuable comments and suggestions.

1. While proposing the mechanism (line 248 on page 10), the authors say "SenB involves the amino side chain of K158 capturing a proton from a water molecule, becoming a negatively charged nucleophile that attacks the positively charged phosphorus atom in the Se-P bond...". There are problems with this sentence and the related figure (Fig. 5f). First, it was the water molecule that becomes negatively charged, not SenB; second, the phosphorus atom here is only "partially positively charged" due to the unequal polarities between P and Se; third, there are several problems with in the "C-Se bond formation" and "Se-P cleavage" steps of the Fig. 5f cartoon, where the arrows were pointing to the wrong atoms or positions;

Response:

Thank you for your valuable comments. We have corrected the sentence to "Thus, the Se-P bond cleavage mediated by K158 in SenB may begin with the amino side chain of K158, which captures a proton from a water molecule. The resulting negatively charged water molecules attack the partially positively charged phosphorus atom in the Se-P bond, leading to bond disruption and the formation of a selenosugar product." In addition, the Fig. 5f has been corrected in the revised manuscript.

2. Site-Directed Mutagenesis in the Methods section: the quickchange method should be clearly stated;

Response:

Thank you for your valuable comments. Description of the Site-Directed Mutagenesis method has been added in the revised manuscript on page 14 line 351-363.

3. The manuscript could've used some language editing; typos or grammatical errors are seen. For instance,

a. line 222 on page 9 "...deploy different strategy": missing "a"

b. line 379 on page 15 "...Tri-HCl"?

c. line 381 on page 15 "...a 6 h incubation period": missing a hyphen;

Response:

Thank you for your valuable comments. "...deploy different strategy" has been corrected to "...deploy a different strategy"; "...Tri-HCl" has been corrected to "...Tris-HCl"; "...a 6 h incubation period" has been corrected to "...a 6-h incubation period". We have corrected the English usage of the whole document through the Nature Research Editing Service.

Reviewer #3 (Remarks to the Author):

I already wrote in my previous review of this paper about the relevance and value of this work; I stand by it: it's a valuable addition for selenium research. Regarding the revision:

Response:

Thank you for your valuable comments.

The minor issues I raised were addressed.

I still recommend improvements in English language throughout the whole document; perhaps authors can use a professional service? Or the journal can step in to help?

Response:

We have corrected the English usage of the whole document through the Nature Research Editing Service.

Reviewer #1 (Remarks to the Author):

This is the second revision of a paper that describes the biochemical and structural characterization of a novel glycosyltransferase involved in the synthesis of a Se-containing metabolite in a broad collection of bacteria. The initial and revised submissions had numerous concerns regarding the writing and interpretation of the experimental data. This revised submission addressed all of these prior concerns and is exceptionally improved. It is presently acceptable for publication and will be a wonderful addition to the seleno-chemistry literature.

One exceptionally minor point that the authors should consider is a revision to Fig. 3b. The linear diagram at the top of Fig. 3b shows the N-terminal domain in blue and C-term domain in green box format but the highlighting boxes in the bottom of the same panel have coloring that is the opposite. Revision of this figure to match the coloring between the top and bottom of the panel would eliminate potential confusion for the reader.

Reviewer #2 (Remarks to the Author):

The authors have successfully addressed all the concerns from this reviewer.

Point-by-point response

REVIEWER COMMENTS

Reviewer #1 (Remarks to the Author):

This is the second revision of a paper that describes the biochemical and structural characterization of a novel glycosyltransferase involved in the synthesis of a Se-containing metabolite in a broad collection of bacteria. The initial and revised submissions had numerous concerns regarding the writing and interpretation of the experimental data. This revised submission addressed all of these prior concerns and is exceptionally improved. It is presently acceptable for publication and will be a wonderful addition to the seleno-chemistry literature.

Response:

Thank you for your valuable comments.

One exceptionally minor point that the authors should consider is a revision to Fig. 3b. The linear diagram at the top of Fig. 3b shows the N-terminal domain in blue and C-term domain in green box format but the highlighting boxes in the bottom of the same panel have coloring that is the opposite. Revision of this figure to match the coloring between the top and bottom of the panel would eliminate potential confusion for the reader.

Response:

We greatly appreciate the reviewer for valuable comments and corrections. We have corrected the Fig. 3b in the revised manuscript.

Reviewer #2 (Remarks to the Author):

The authors have successfully addressed all the concerns from this reviewer.

Response:

Thank you for your valuable comments.